# Magnetoelectric effect in multiferroic nickelate perovskite YNiO$_3$
Nazaret Ortiz Hernández [1,3], Elizabeth Skoropata [1,3], Hiroki Ueda [1], Max Burian[1], José Antonio Alonso [2] & Urs Staub [1] ✉

The interaction of magnetic order and spontaneous polarization is a fundamental coupling with the prospect for the control of electronic properties and magnetism. The connection among magnetic order, charge localization and associated metal-insulator transition (MIT) are cornerstones for materials control. Materials that combine both effects are therefore of great interest for testing models that claim the occurrence of spontaneous polarization from magnetic and charge order. One class of materials proposed to combine these functionalities is the family of RNiO$_3$ (R: Lanthanide or Yttrium), whose members show a clear MIT and an antiferromagnetic ground state and for which an electric polarization has been predicted. Here, using resonant magnetic x-ray scattering with circular polarization and an applied electric field we show that YNiO$_3$ possess a magnetic structure containing domains of spin-rotations that are consistent with an electric polarization. We show a reversal of the magnetic structure with the applied electric field confirming that charge ordered RNiO$_3$ are magnetoelectric type II multiferroics with a MIT.

Magnetoelectric materials, and in particular the subset of materials with coexisting magnetic order with spontaneous electric polarization (magnetoelectric multiferroics), have attracted great interest in the last two decades. This has been motivated by the possibilities of using conjugate fields for magnetization and polarization switching in view of possible novel applications[1]. The base of the underlying magnetoelectric effect is the symmetry of the crystal structure and that of the magnetic structure. If a magnetic structure is incompatible with the symmetry elements for the crystal structure, then the onset of the magnetic order lowers the crystal symmetry of the material. This is the case for magnetically-driven type II multiferroics, in which an electric polarization appears at the magnetic phase transition and the atoms move from their higher symmetry positions due to a loss of inversion symmetry driven by the magnetic order. The atomic displacements created depend on the electron-phonon and spin-lattice interactions and can be too small to be quantified using standard x-ray or neutron diffraction methods. In fact, the typical electric polarization amplitude in multiferroics is three orders of magnitude smaller than that in ferroelectrics. Even for prototypical type II multiferroics such as TbMnO$_3$[2] the exact structure in the multiferroic state is not experimentally available, even though indirect experimental methods found some extremely small distortions[3].

As the build-up of an electric polarization is hindered in conductive materials, it is not straightforward to test if such materials exhibit an intrinsic polarization. This can be applied to materials that have a metal-insulator transition (MIT), as these materials often have measurable resistances (rest conductivity) even deep in the insulating state. So far, only in the classical charge-ordered magnetite (Fe$_3$O$_4$), a polarization could be found far below the MIT temperature[4]. An electric polarization has been predicted to occur from charge ordering in manganites[5] and has also been applied to the RNiO$_3$ family (R being a Lanthanide or Y)[6]. It has been shown by calculations that different magnetic structures lead to different types of multiferroicity in the latter series[7]. Recent theoretical work also described multiferroicity in nickelates, however, did not consider the non-collinear magnetic structure[8]. No polarization measurements have been reported in these nickelates, as this is often not possible in slightly conductive materials. Indirect experimental methods can help to test the loss of space inversion symmetry; for example, a recent Raman study[9] observed additional phonon modes below the magnetic phase transition temperature, supporting a loss of inversion symmetry, but not all magnetic non-centrosymmetric materials are multiferroic, e.g., the proper screw phase in a Y-type hexaferrite. A direct detection of changes in the magnetic response by the application of an electric field is proof of the magnetoelectric effect and forms the base of multiferroic applications. However, such evidence remains elusive in nickelates, despite their growing relevance with the discovery of superconductivity[10]. Since these exotic physics, i.e., multiferroicity, superconductivity, and MIT, are all based on the strong connection among electrons, lattice, and spins, insights gained from a demonstration of magnetoelectric coupling in a new class of materials can point to new pathways to control correlated phenomena.

[1]Swiss Light Source, Paul Scherrer Institute, Forschungssrtasse 111, 5232 Villigen-PSI, Villigen, Switzerland. [2]Instituto de Ciencia de Materiales de Madrid, CSIC, Cantoblanco, E-28049 Madrid, Spain. [3]These authors contributed equally: Nazaret Ortiz Hernández, Elizabeth Skoropata. ✉e-mail: urs.staub@psi.ch

The RNiO$_3$ series exhibits a MIT with transition temperatures strongly varying with the R ion size[11]. For R=La the material remains metallic at all temperatures, whereas for a smaller ionic radius of the R ions, a higher MIT temperature is found. At high temperatures, the crystal symmetry is described in Pbnm with a single Ni site. At temperatures lower than the MIT temperature, the symmetry is reduced to P2$_1$/n[12] with two distinct Ni sites (Ni I and Ni II) that have different Ni-O bond lengths. This symmetry reduction is interpreted as the onset of charge order at the Ni[12] as supported by resonant x-ray diffraction[13]. More recently, the bond length separation with an electronic state ordering of $d^8\underline{L}^2$ and $d^8$ states with holes at the oxygen was used to describe the MIT[14,15]. In the insulating state, RNiO$_3$ exhibit an antiferromagnetic order with a (½ 0 ½) wave vector that has first been assigned to an up-up-down-down collinear spin structure[16] and has later been shown to be a non-collinear magnetic structure by soft x-ray magnetic scattering in epitaxially grown thin films[17]. Further resonant soft x-ray diffraction has been extensively used to investigate the magnetic properties of films[18–22]. In addition, it has been shown theoretically that both magnetic structures break inversion symmetry[7] and the polarization will depend on the choice of the magnetic ground state, suggesting the existence of a magnetoelectric coupling.

Here we directly address the magnetoelectric effect by studying the magnetic structure in applied electric fields of polycrystalline bulk YNiO$_3$ samples, for which the MIT temperature is much higher than that of the antiferromagnetic magnetic order[12] and the sample is a good insulator in the magnetically ordered state. Polycrystalline samples allow us to investigate the intrinsic ground state of YNiO$_3$ where the properties are unaffected by strain as in thin film systems, and where the grains are switchable with a reasonable applied electric field. Using circular x-ray polarization, we can directly probe the existence of a "spiral" non-collinear magnetic structure and its handedness that creates circular dichroism in the magnetic diffraction peaks. In addition, a change in x-ray polarization contrast after reversing the applied electric field represents a magnetoelectric coupling verifying the multiferroicity of the material.

## Results and discussion

The magnetic structure of polycrystalline RNiO$_3$ has already been investigated previously[23,24] with resonant soft X-ray diffraction that results in magnetic scattering occurring in a Debye Scherrer ring. Experiments are performed with the RESOXS endstation[25] connected to the SIM beamline[26] at the Swiss Light Source located at the Paul Scherrer Institute. The total scattering angle of the magnetic (½ 0 ½) reflection is at approximately 117° for x-ray energies around 853 eV (Ni $L_3$ edge) that is covered by an in-vacuum charge coupled device (CCD) camera located approximately 30 cm from the sample. The CCD camera covers a few degrees of the Debye Scherrer ring of the polycrystalline YNiO$_3$ sample, with the intersected ring being vertical at this angle (see Fig. 1). For more experimental details, see Methods. Due to the large spot size of the unfocused beam (~2 × 2 mm), the sampling rate of polycrystals that fulfill the Bragg scattering conditions allows the magnetic signal of several grains to be collected simultaneously even when the penetration depth at resonance (in the order of 100 nm) is much smaller than the grain size of several microns.

The proposed 90 degrees non-collinear magnetic structure composes two interpenetrating collinear AFM sublattices formed by Ni I and Ni II magnetic sites, respectively. As the spins of one sublattice are all perpendicular to those of the other (see Fig. 2), the effective magnetic structure shows spin rotations along all former cubic perovskite axes (with all nearest neighbor spins being perpendicularly oriented), which correspond to an orthorhombic structure to (110)$_o$, (1–10)$_o$ and (001)$_o$. This is in contrast to a cycloid e.g. in TbMnO$_3$ where the spins rotate only along the wave vector **q**[2]. The scattering intensity from a single domain state of a cycloid will show circular dichroic contrast in the magnetic diffraction signal as observed in the cycloidal order of TbMnO$_3$[27,28]. The dichroic contrast will be strongly dependent on the azimuthal angle defined by the angle between the cycloid plane and the scattering plane. Here we first like to address how the rotations of the spins along the (110)$_o$, (1–10)$_o$ and (001)$_o$ directions result in circular contrast in resonant magnetic scattering that depends on the domain similar to a cycloid. The magnetic intensity for circular polarization can be obtained

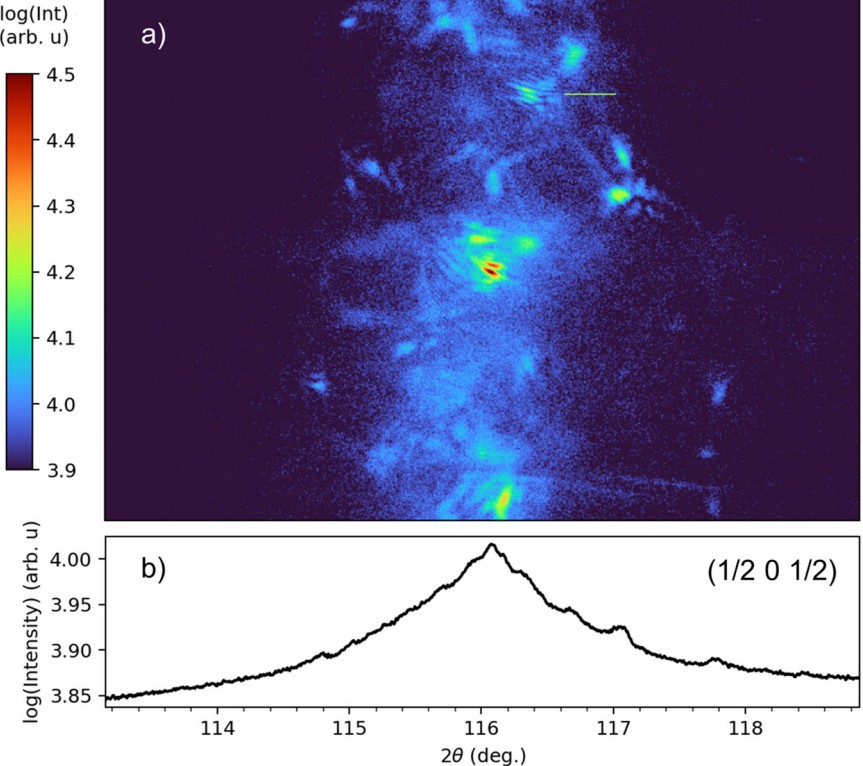

**Fig. 1 | Resonant magnetic x-ray scattering from polycrystalline YNiO$_3$.** Measurement is taken at 50 K after cooling from 200 K (>$T_N$) in an E-field with 5 kVcm$^{-1}$. **a** Image from the 2D detector showing contributions from different grains, taken with $C+$ polarization at 854.6 eV and with an unfocused beam approximately 2 × 2 mm in size. **b** Vertical projection of the scattering intensity as a function of 2θ.

from the structure factors of linear polarization as expressed in ref. 29

$$I_x = \left|F_{\pi'\pi}\right|^2 + \left|F_{\sigma'\pi}\right|^2 + \left|F_{\pi'\sigma}\right|^2 - \chi \text{Im}\left\{F_{\pi'\pi}^* F_{\pi'\sigma}\right\} \quad (1)$$

with $\chi$ being the degree of circular light polarization being ±1 for left- and right-handed polarization. The circular light dependence occurs when the complex phases of the two structure factors of the linear polarization channels, $F_{\pi'\sigma}$ and $F_{\pi'\pi}$ are out of phase, which is only achievable with a non-collinear spin structure[30]. The magnetic structure factor for the (½ 0 ½) reflection for the interpenetrating collinear magnetic structure can be decomposed, keeping the same origin, to

$$F^{\text{mag}} = F_{\text{NiI}}^{\text{mag}} + i F_{\text{NiII}}^{\text{mag}}, \quad (2)$$

as the two sites have a structural coordinate difference modulus 1/4 of the cell in the structure factor for this reflection, and both $F_{\text{NiI}}^{\text{mag}}$ and $F_{\text{NiII}}^{\text{mag}}$ being

pure imaginary. We now consider the polarization dependence of magnetic scattering at resonance[31] and assume a perfect 90 degree moment direction difference in the ac plane for the two sites. The cross product $F_{\pi'\pi}^* F_{\pi'\sigma}$ becomes purely imaginary (maximal circular dichroic contrast) if the moments of one sublattice are perpendicular to the scattering plane spanned by the incoming and outgoing x-ray wavevectors **k** and **k'** and the others are parallel to the ordering wave vector (**k-k'** // **q**), resulting in the b-axis being // **k + k'** along which there is no magnetic moment component. For **b** ⊥(**k + k'**) and ⊥(**k-k'**), i.e., b perpendicular to the scattering plane, the circular contrast in the magnetic scattering is absent. Thus, it gives rise to azimuthal angle dependent dichroism as for a magnetic cycloid. Opposite domains exist and simply consist of reversing all spins of one of the sublattices, resulting in a sign change of the dichroism as one of the structure factors changes the sign, which reverses the handedness of the spin rotations along all three directions. Figure 3a shows the magnetic diffraction response obtained with a focused beam of approximately $150 \times 150$ µm that selects just a few grains. Figure 3b, c shows the rocking curve of the region of interest of the upmost and lowest peaks of the detector image for opposite circular polarized x-rays. A clear and opposite dichroism is observed between the two peaks (Fig. 3d), which indicates two grains that have approximately opposite orientations of the b axis. Note that the (101) direction is fixed by the Bragg condition. The observed circular light contrast directly shows that the magnetic structure must be non-collinear also in bulk (polycrystalline) YNiO₃, supporting that bulk YNiO₃ should indeed be considered a multiferroic with a magnetic structure breaking space inversion symmetry[32].

It is important not only to show that the magnetic order intrinsically breaks inversion symmetry but also to show a magnetoelectric effect. The most direct way is to display that a magnetic domain can be switched by an applied electric field. In the most simple model for creating a polarization from the magnetic structure, the spin current model, the direction of the polarization **P** is given by **P** $\propto$ **r**$_{ij}$ × (**S**$_i$ × **S**$_j$)[33], with **r**$_{ij}$ being the unit vector between neighboring spins **S**$_i$ and **S**$_j$. We can consider the helicity of the nearest neighbor spins along the **r**$_{ij}$ // $[110]_o,[1{-}10]_o$ and $[001]_o$ axis separately. The term (**S**$_i$ × **S**$_j$) is for all neighbor pairs the same and results in a residual vector being parallel to the b axis. For the two in-plane directions, the polarization will be along $[001]_o$, reduced due to the non-perpendicular directions of the b axis to **r**$_{ij}$ because the crystal structure is orthorhombic. For the $[001]_o$ direction, **P** // $[-1\ 0\ 0]_o$, resulting in a net polarization that has a strong component along the $[-1\ 0\ 1]_o$ direction. As opposite domains have a reversed helicity along all the directions, there is a sign change of **S**$_i$ × **S**$_j$, which results in a reversal **P**. Note that it is possible that the easy axis of the two sites is opposite (it is defined by the low site symmetry creating defined site magnetic anisotropy, which is not experimentally known) and would result in 90 degree rotation of all spins, which would not change the

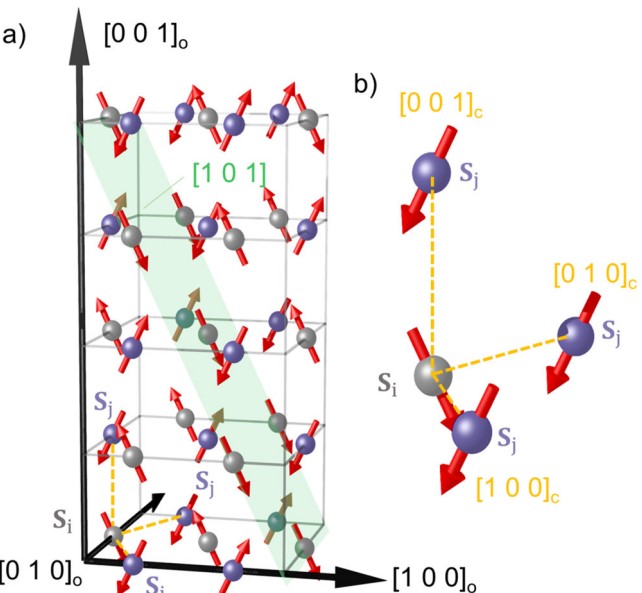

**Fig. 2 | Sketch of magnetic structure of YNiO₃. a** It shows only Ni I (purple sphere) and Ni II (gray sphere) atoms with spin rotations along the former cubic a–c directions. Along nearest neighbor bonds, spins rotate along a single handedness. **b** nearest neighbour spins along the principal axis of cubic symmetry.

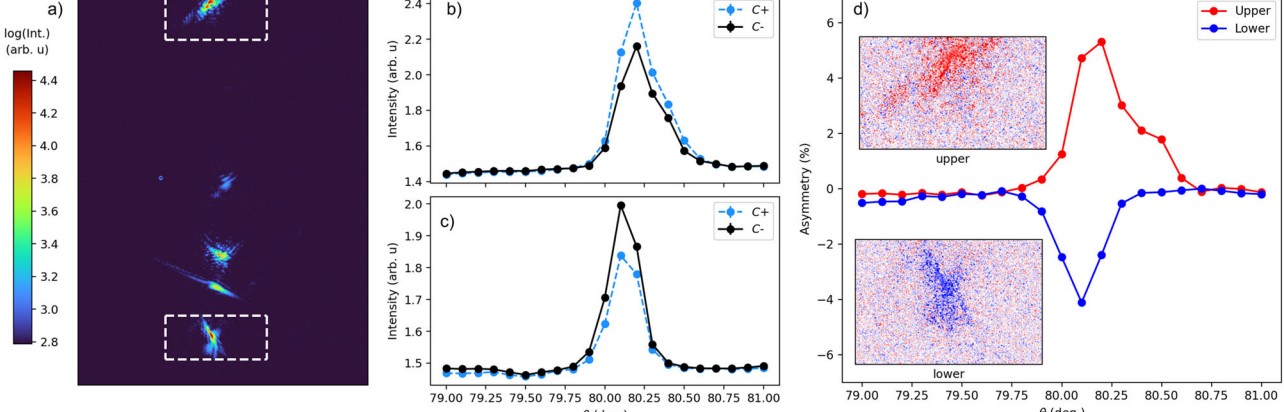

**Fig. 3 | Circular polarization dependence of magnetic reflections. a** Magnetic (½ 0 ½) reflections measured at 855.0 eV (first Ni L₃-edge peak) and 10 K from a polycrystalline sample. Circular polarization contrast in the rocking (θ) curves for integrated intensities of different grains creating the (**b**) upmost and (**c**) lowest

reflections visible on the image (3a). **d** Rocking curve of the asymmetry $[I(C+) - I(C-)]/[I(C+) + I(C-)]$ of the reflections of these two grains, with the insets showing the polarization contrast in the detector images at the maxima of the rocking curve (θ = 80.2 deg). Statistical errors (SD) are smaller than the markers.

**Fig. 4 | Electric field dependence of magnetic reflections.** X-ray circular light contrast of the magnetic (½ 0 ½) reflection at 855.0 eV (first Ni $L_3$-edge peak) and 10 K for individual grains (**a**) after field cooling with $-10\,kVcm^{-1}$ and (**c**) after iso-thermal field switching to $+10\,kVcm^{-1}$. **b** and **d** show the integrals of the reflection intensities of individual exposures together with an integral of the background region. A clear magnetoelectric effect is observed in the reflection intensities, but not in the background. The images also show clear effects due to a coherent fraction of the x-ray beam hitting the individual grains. However, reconstructions did not result in stable domain/particle images due to missing boundary conditions. Statistical errors (SD) are smaller than the markers.

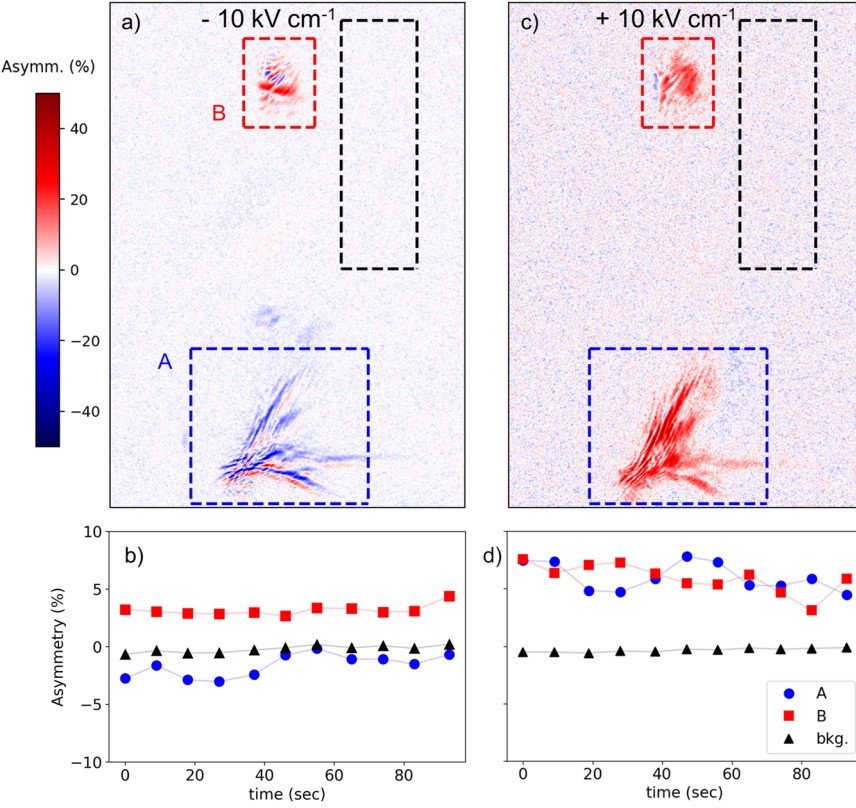

above considerations. Important here is that in the presence of a magne-toelectric coupling, a reversal of **P** reverses the spin rotations along the three directions, leading to opposite circular dichroism in the magnetic diffraction signal.

To investigate the resulting magnetoelectric effect, we use a small pellet of pressed (sintered) powder in a capacitor like structure with a 1 mm gap. For the measurement in applied electric fields, we choose an incident angle and spatial position where we can observe two magnetic reflections simultaneously originating from domains (grains) with opposite signs of dichroism (see Fig. 4). This allows to make sure that the observed dichroism is not caused from a normalization problem. We field-cooled the sample through antiferromagnetic order temperature in $-10\,kVcm^{-1}$ applied field and kept the field on while taking the dif-fraction images (Fig. 4a) and then reversed the applied field at low temperatures to $+10\,kVcm^{-1}$ and repeated the measurement (Fig. 4b). The corresponding circular light dichroism for field cooling followed by the isothermal field reversal is shown in Fig. 4a, b and results in opposite dichroic contrast of one of the domains. This is a direct indication of the magnetoelectric effect, which shows that $YNiO_3$ is indeed a multiferroic. Unfortunately, a stable switching of multiple domains could not be achieved due to the strong effect of X-rays creating photoelectrons leading to charging (and discharges). This can affect the polarization after longer exposure times with x-rays following the initial field-cooling and cycling as e.g. has been seen in $DyMnO_3$[34]. Random switching of multiferroic domains probed by pulses of X-rays has also recently been observed in films of $Ge_{1-x}Mn_xTe$ and its switching behavior has been studied in detail for intermittent exposure (measuring) times[35]. Also in our measurements, the x-ray exposure is pulsed, as the beam is blocked during the readout time of the CCD camera. Even though one of the domains did not flip its contrast by the field cooling, the amplitude of the contrast is enhanced by the in-situ electric-field switching. This suggests that the peak is consisted of reflections from multiple domains with different in-plane orientation and that some of the domains switch their spin-spiral direction because of a large enough electric field projection along their polar axis.

We like to comment on the shape of the diffraction peaks, which have interesting speckle structures typical for coherent (magnetic) Bragg dif-fraction. Speckles can be observed as the optical elements of the beamline image the source onto the sample. As the grains are much smaller than the x-ray beam focus, the grain selects a fraction of coherent x-rays from the source. This is equivalent to using a pinhole in the beamline to select a spatially coherent fraction of x-rays. Unfortunately, reconstructions of the speckles from the grains did not converge and we could not obtain a stable real space image of the magnetic domains. Nevertheless, the data might show a path for coherent diffraction imaging when smaller particles are available. Besides, the speckle pattern supports the presence of multiple domains in a single grain, as we discussed above.

It has been shown theoretically that also a collinear up-up-down-down magnetic structure would result in an electric polarization[7] connected to two different magnetic domain states selectable by an electric field. However, in that case, the individual domains would not result in a dichroic effect on the magnetic diffraction peak because the spins are collinear. Without being sensitive to the phase of the magnetic scattering, standard resonant x-ray diffraction cannot distinguish such domains. The observation of a non-collinear magnetic structure has a further implication on the occurrence of a fractional charge order at the Ni[12,13] or the oxygen sites[14,15] with the latter being currently the preferred interpretation[36]. Note that a separation into $d^8$ and $d^8\underline{L}^2$ electronic configuration, with $d^8\underline{L}^2$ being a state with $S = 0$, cannot result in a non-collinear magnetic structure from Ni magnetic moments sampled in our experiment. Our experimental findings require a significant non-zero moment on both sites, which needs to be taken into account for a microscopic description of the electronic Ni and oxygen states.

## Conclusions
In summary, resonant soft x-ray diffraction experiments on the magnetic (½ 0 ½) reflection showed clear circular dichroic contrast indicative of a non-collinear magnetic structure of bulk $YNiO_3$. This shows that bulk and strained films have the same magnetic structure. A clear sign change of the dichroic signal has been observed in one of the grains when reversing the

**Article**

electric fields, representing a magnetoelectric effect confirming that $RNiO_3$ perovskites are multiferroic in its bulk form. These results might open up the possibilities for turning on the magnetoelectric effect in the $RNiO_3$ nickelates with lighter R elements for which the MIT and multiferroicity occur simultaneously, possibly through overcoming the finite resistance at the transition by the use of ultrashort electric field pulses.

## Methods

$YNiO_3$ compounds were prepared as polycrystalline powder by high-pressure solid-state reactions, as described in the literature[23]. It is the same sample as already used in ref. [24]. Resonant soft x-ray magnetic diffraction experiments were performed with the RESOXS endstation[25] at the SIM beamline of the Swiss Light Source of the Paul Scherrer Institute, Switzerland. Polycrystalline pellets of 10 mm diameter were glued onto a copper sample holder mounted on a He flow cryostat, which achieves temperatures between 10 and 370 K. For the electric field experiments, a pellet strip of 1 mm size was mounted between two copper electrodes and mounted on the sapphire plate, allowing the application of a large voltage on the sample in this condenser-like geometry. For that part the sample was field cooled and then the field reversed to look in-situ to switch the polarization of the material. X-ray experiments were performed using circularly left/right polarized x-ray beams. Two-dimensional data sets were collected with a commercial Roper Scientific charge couple device (CCD camera) mounted in the vacuum chamber. The recorded CCD images were corrected for background (recorded at the same energy but with $2\theta = 5$ degrees offset), integrated along the camera height (i.e. along the Debye–Scherrer ring) and fitted with a Lorentzian and a linear background function. This linear background originates mainly from the fluorescence of the sample.

## Data availability

Experimental data are accessible from the PSI Public Data Repository (https://doi.org/10.16907/d18dccdd-e03f-4494-afe8-1ffe045267de).

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

## Acknowledgements

We like to acknowledge Valerio Scagnoli for stimulating discussions and the X11MA beamline staff for experimental support. Experiments have been performed at the X11MA beamline of the Swiss Light Source at the Paul Scherrer Institute under Proposal No. 0190537. N.O.H. and M.B. acknowledge financial support of the Swiss National Science Foundation, No. 200021_169017. H.U. was supported by the National Centers of Competence in Research in Molecular Ultrafast Science and Technology (NCCR MUST No. 51NF40-183615). E.S. received funding from the European Union's Horizon 2020 research and innovation programme under the Marie Skłodowska-Curie grant agreement No 884104 (PSI-FELLOW-III-3i). J.A.A. thanks the Spanish Ministry for Science and Innovation (MCIN/AEI/10.13039/501100011033) with grant numbers: PID2021-122477OB-I00.

## Author contributions

N.O.H. and U.S. conceived and designed the project. J.A.A. made the $YNiO_3$ sample. N.O.H., H.U., M.B. and U.S. performed the resonant magnetic soft X-ray diffraction experiment, E.S. analyzed the experimental data with inputs from N.O.H. and U.S. Finally, E.S., H.U. and U.S. interpreted the results and wrote the manuscript with contributions from all authors.

## Competing interests

The authors declare no competing interests.
