## [Peer Review File · Communications Materials]

This manuscript has been previously reviewed at another Nature Portfolio journal. This document only contains reviewer comments and rebuttal letters for versions considered at Communications Materials.

25th Jul 24

Dear Dr Staub,

Your manuscript titled "Magnetoelectric effect in multiferroic nickelate perovskite YNiO₃" has now been seen again by our referees, whose comments appear below. In light of their advice I am delighted to say that we are happy, in principle, to publish a suitably revised version in Communications Materials.

We therefore invite you to revise your paper one last time to address the remaining suggestion of Reviewer #1. At the same time we ask that you edit your manuscript to comply with our journal policies and formatting style in order to maximise the accessibility and therefore the impact of your work.

EDITORIAL REQUESTS

* Your manuscript should comply with our policies and format requirements, detailed in our style and formatting guide (<https://www.nature.com/documents/commsj-phys-style-formatting-guide-accept.pdf>).

* Please edit your manuscript according to the editorial requests in the attached table, and outline revisions made in the right hand column. If you have any questions or concerns about any of our requests, please do not hesitate to contact me. It is important that each request be addressed in order to avoid delays in accepting your manuscript. Please upload the completed table with your manuscript files as a Related Manuscript file.

* The editorial requests table also includes a full list of the files that must be provided upon resubmission. Please upload your files according to this table.

* An updated editorial policy checklist that verifies compliance with all required editorial policies must be completed and uploaded with the revised manuscript. All points on the policy checklist must be addressed; if needed, please revise your manuscript in response to these points. Please note that this form is a dynamic 'smart pdf' and must therefore be downloaded and completed in Adobe Reader. Clicking this link will download a zip file containing the pdf.

OPEN ACCESS

Communications Materials is a fully open access journal. Articles are made freely accessible on publication. For further information about article processing charges, open access funding, and advice and support from Nature Research, please visit <https://www.nature.com/commsmat/open-access>

Please use the following link to submit your revised files:

[link redacted]

We hope to hear from you within two weeks; please let us know if the process may take longer.

Best regards,

Aldo

Dr Aldo Isidori

Senior Editor

Communications Materials

REVIEWERS' COMMENTS:

Reviewer #1 (Remarks to the Author):

During the response, the authors explained and addressed the comments from the referees and made further revision of the manuscript. With this, the referee tends to suggest the publication of this work in communications materials with the following suggestion.

Comment: The essential result of this work is the demonstration of switched image contrast through electric field, which is explained as the change of handedness/polarity by the authors. However, it might be possible that the orientation of the grain is modified with the electric field through the the lattice-electric field interactions, such as electrostriction. The referee would suggest the authors to elaborate this issue in the manuscript.

Reviewer #3 (Remarks to the Author):

The authors have appropriately replied to most of the reviewers' concerns.

I recommend publication of this work in its current form.